# The role of risk communication in public health interventions. An analysis of risk communication for a community quarantine in Germany to curb the SARS-CoV-2 pandemic

Juliane Scholz[1]*, Wibke Wetzker[1], Annika Licht[1], Rainer Heintzmann[2], André Scherag[3], Sebastian Weis[1,4], Mathias Pletz[4], Cornelia Betsch[5], Michael Bauer[1], Petra Dickmann[1], the CoNAN study group[¶]

1 Department of Anaesthesiology and Intensive Care Medicine, Jena University Hospital, Friedrich Schiller University, Jena, Germany, 2 Leibniz Institute of Photonic Technology, Institute of Physical Chemistry and Abbe Center of Photonics, Friedrich Schiller University, Jena, Germany, 3 Institute of Medical Statistics, Computer and Data Sciences, Jena University Hospital, Friedrich Schiller University, Jena, Germany, 4 Institute for Infectious Diseases and Infection Control, Jena University Hospital, Friedrich Schiller University, Jena, Germany, 5 Media and Communication Science, University of Erfurt, Erfurt, Germany

¶ Membership of the CoNAN study group is provided in the Acknowledgements.
* j.scholz@uni-jena.de

**Data Availability Statement:** All relevant data are within the manuscript and its Supporting information files.

## Abstract

### Background

Separating ill or possibly infectious people from their healthy community is one of the core principles of non-pharmaceutical interventions. However, there is scarce evidence on how to successfully implement quarantine orders. We investigated a community quarantine for an entire village in Germany (Neustadt am Rennsteig, March 2020) with the aim of better understanding the successful implementation of quarantine measures.

### Methods

This cross-sectional survey was conducted in Neustadt am Rennsteig six weeks after the end of a 14-day mandatory community quarantine. The sample size consisted of 562 adults (64% of the community), and the response rate was 295 adults, or 52% (33% of the community).

### Findings

National television was reported as the most important channel of information. Contact with local authorities was very limited, and partners or spouses played a more important role in sharing information. Generally, the self-reported information level was judged to be good (211/289 [73.0%]). The majority of participants (212/289 [73.4%]) approved of the quarantine, and the reported compliance was 217/289 (75.1%). A self-reported higher level of concern as well as a higher level of information correlated positively with both a greater acceptance of quarantine and self-reported compliant behaviour.

**Funding:** PD received the awards #5575/2-1 63952/2020 and #5526/32-4-2. Thuringian Ministry for Economic Affairs, Science and Digital Society URL: https://wirtschaft.thueringen.de/ The funders had no role in study design, data collection and analysis, decision to publish, or preparation of the manuscript.

**Competing interests:** The authors have declared that no competing interests exist.

## Interpretation

The community quarantine presented a rare opportunity to investigate a public health intervention for an entire community. In order to improve the implementation of public health interventions, public health risk communication activities should be intensified to increase both the information level (potentially leading to better compliance with community quarantine) and the communication level (to facilitate rapport and trust between public health authorities and their communities).

## Introduction

The 2020 pandemic caused by severe acute respiratory syndrome coronavirus-2 (SARS-CoV-2) presents an exceptional challenge to the global community. As of June 2021, no causal therapy exists and even with the now approved vaccines, it is a challenge to reach herd immunity due to vaccine hesitancy, limited production capacity and uncertainty regarding the duration of vaccine protection [1]. Therefore, non-pharmaceutical interventions are important and effective measures to mitigate the spread of the virus and to limit the pandemic's impact on societies. The most effective non-pharmaceutical intervention to interrupt chains of transmission within communities is the separation of ill (isolation) or possibly infectious (quarantine) persons from non-infected communities. Quarantine can be applied at the level of the individual, group or community (*e.g.* entire villages). Interventions that aim to minimise transmission rates by impeding what is perceived as "normal" public life are commonly referred to as "lockdown" [2–4].

Public health authorities play a pivotal role in implementing public health interventions. Risk communication activities can modify influencing factors for the successful implementation of such measures.

Evidence shows that effective risk communication strategies emphasise the role of information, communication and coordination as risk governance of health authorities [5]. In addition, building a relationship with the affected communities is important to foster rapport and trust [6]. Importantly, the level of trust in public health authorities and the government can positively influence the acceptance of measures [7].

While authorities apply risk communication strategies in order to implement public health interventions, one of the indicators of successful implementation is the compliance of and within the community. An important factor influencing the compliance of individuals, communities and societies is the *fear* of contagion [8]. Earlier studies revealed that level of concern, irrespective of actual exposure, is a driver for health information seeking behaviour [9]. Risk communication therefore plays an important role in risk evaluation and adoption of preventive behaviours. Another important driver for compliance is the wish to protect one's family members [10]. Studies show that compliance at the community level is greatest when the affected community understands the reasons for such measures and trusts their relevance and balance [10]. Further factors influencing acceptance at the societal level are the existing social norms and the perceived benefits of quarantine for society [11].

Yet there are scarce data on *how* to successfully implement quarantine. Current research in the area of quarantine focuses on its psychological impact on individuals or on epidemiological aspects of disease transmission [12,13]. This is also true for recent German studies of SARS-CoV-2 hotspots–of which some had been placed under community quarantine–that focused

on epidemiological aspects, including antibodies and viral load assessments in various settings [14–16]. None of these studies investigated the implementation of a community quarantine.

Community quarantine differs from individual quarantine because it does not represent a linear logic between illness and intervention but establishes the link between an assumed exposure with a likelihood of infection and the intervention for an entire group [17,18]. Therefore, more and different communication is required [19]. The aim of our study was to identify conditions and influencing factors that facilitate risk communication governance, flow of information, communication and coordination.

## Background

In Neustadt am Rennsteig, a cluster of six infections could no longer be contact traced, which represents a key measure of disease control and successful pandemic management [20]. Therefore, one of the first community quarantines in the context of the SARS-CoV-2 pandemic in Germany was imposed on the village from 22 March to 5 April 2020 [21]. During this time period, all 883 inhabitants were placed under quarantine [22].

Fig 1 shows that the quarantine ended with a total of 47 confirmed cases, identifying both symptomatic and asymptomatic patients but excluding fatalities. Overall, 51 SARS-CoV-2 infections were confirmed during the outbreak, including three fatalities [22]. That figure represents 5·8% of Neustadt's inhabitants, compared to 0·05% of confirmed cases in Thuringia

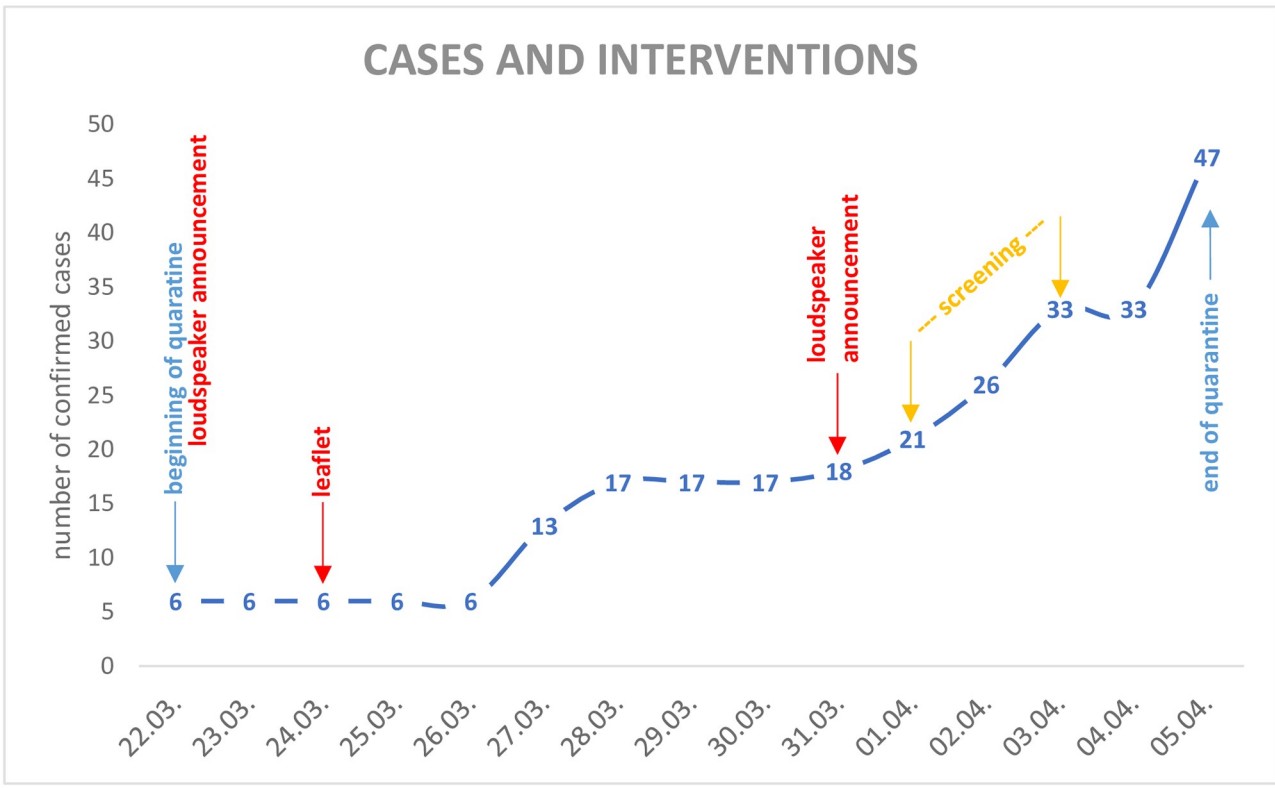

**Fig 1. Overview of the confirmed cases of SARS-CoV-2 in Neustadt am Rennsteig and the interventions of the local authorities.** The beginning of the community quarantine was announced via loudspeaker as well as the screening of every inhabitant (symptomatic or asymptomatic) in the second week. The third day of quarantine (24.03.2020) a leaflet was handed out with all the necessary information. You can find the issued order and the leaflet in the Supplementary Material. [number of confirmed cases according to https://www.ilm-kreis.de/Landkreis/Ver%C3%B6ffentlichungen/Pressearchiv/index.php?ModID=255&object=tx%2C2778.5.1&La=1&NavID=2778.25&text=&kat=&monat=202003&jahr= (accessed: 28.04.2020)].

and 0·11% of confirmed cases in Germany at the time [23]. By screening all inhabitants of Neustadt during the second week of quarantine, all remaining active cases could be isolated and further transmission could be prevented.

The quarantine was announced with immediate effect on a Sunday evening by loudspeaker announcement followed by an information leaflet distributed the following Tuesday (day 2 of quarantine), and information was posted on the municipality's website. The loudspeaker announcement was repeated the following Tuesday (day 9 of quarantine), emphasising the rules of quarantine and inviting residents to be screened.

The local health authorities were responsible for the implementation of and communication during community quarantine. They did not follow a pre-designed protocol but drafted an ad-hoc plan, changing their strategy day by day, adapting to upcoming issues.

The investigation of the risk communication during quarantine took place six weeks after the intervention had ended.

## Methods

### Study design and participants

The study employed a cross-sectional survey conducted over three days during the administration of the CoNAN study [22] (13–16 May 2020) in Neustadt am Rennsteig. The retrospective study took place six weeks after the mandatory 14-day quarantine for the entire community had ended.

Our risk communication study was undertaken in addition to the CoNAN seroprevalence study researching the same population, using the same study infrastructure (distributing and collecting questionnaires).

Study participation was anonymous and voluntary. Participants' eligibility was determined based on their age (over 18 years old), residency in Neustadt am Rennsteig during quarantine and participation in the CoNAN study.

The targeted total sample size was the adult population of the community that participated in the CoNAN study (n = 562). A total of 295 participants (52% response rate) returned the questionnaire.

### Ethics approval

The study was conducted according to the current version of the Declaration of Helsinki and has been approved by the institutional ethics committees of the Jena University Hospital, Friedrich Schiller University, and the respective data protection commissioner (approval number 2020–1776) and the ethics committee of the Thuringian chamber of physicians.

The study is registered at the German Clinical Trials Register: DRKS00022416.

### Procedure

Data were collected using a questionnaire that was developed on the basis of a systematic literature review and previous research by our team members. The questionnaire was piloted by team members, discussed with a group of collaborating psychologists (COSMO study group [24,25]) and revised according to their feedback. Due to the urgent need for scientific information accompanied by time constraints at the beginning of the pandemic, there was only limited time to develop a thorough study design and we had to focus on efficiency and speediness. Moreover, due to the small study population, we were reluctant to pilot the questionnaire within the population of Neustadt am Rennsteig as this would mean the loss of a significant portion of potential participants.

The printed questionnaire was handed out together with the epidemiological questionnaire. The distribution of the questionnaire took place before the blood tests for the CoNAN study were performed and the survey was returned afterwards by the participants, giving them enough time to answer the questions. The participants were briefed by the personnel on-site on how to fill out the questionnaire. The two data sets of the CoNAN study and the risk communication study were not connected by participant ID in order to meet the anonymization request of our study.

The full survey is available in the *Supporting Information*. The survey first collected demographic data (age, gender, number of people living in the household) and consisted of items on sources, channels and perceived levels of information regarding the pandemic and quarantine, communication with authorities and within the community and coordination of the quarantine with regards to acceptance, compliance and concerns within the community. The structure of the survey was based on the understanding of risk communication as being composed by three pillars: information, communication and coordination. All three components contribute to capacity building and preparing for the event of a public health crisis [5]. The importance of information in the context of a public health emergency is showcased by the definition of risk communication as "information exchange about health risks caused by environment, industrial, or agricultural, processes, policies, or products among individuals, groups and institutions" [26]. Therefore, the participants' self-reported level of information was one of our focal points. In our study, level of information was defined as the participants' personal grasp of quarantine and the reasons behind the decision which is closely related to the sources and channels of information which were used to access information and which we strived to identify.

The questionnaire consisted of four types of questions: binary closed (6/16), single quantifying (3/16), categorical with an "other" option (5/16) and open-ended (2/16). The closed questions had a five-tier Likert scale answer, and in most cases the results for occasional (3), frequent (4) and very frequent (5) were combined for each answer to gain a clearer understanding of the results.

## Statistical analysis

Descriptive analysis was performed to identify the response characteristics for each questionnaire item and the socio-demographic characteristics of the participants. Absolute and relative frequencies are reported for the binary and categorical response options, whereas semi-structured variables are summarised as mean with standard deviation (SD) and medians with interquartile range. Furthermore, correlation between items was analysed using Spearman's rank correlation. All confidence intervals (CIs) were calculated with 95% coverage. We reported Clopper-Pearson CIs for proportions. The p-values are unadjusted and two-sided. All statistical analyses were performed using SPSS and Excel.

Additional exploratory analyses were performed for open-ended questions; results will be addressed in a separate publication.

The concept of source and channel as used in the following is based on Berlo's model of Source-Message-Channel-Receiver (SMCR). Berlo defines "source" as the origin of the message or the person who originates the message, as for example public health authorities. The source is defined by a number of factors which affect the communication process such as communication skills, knowledge of the subject and attitude towards the audience and the subject. Mediums used to send the message such as telephone, internet or loudspeaker announcements are labelled as channels [27].

## Results

Out of 295 returned questionnaires, six were not valid due to non-completion (less than 50% of the questions answered). A total of 289 valid questionnaires represent 33% of the total population of Neustadt am Rennsteig (n = 883).

### Descriptive analysis

Of the 289 participants, 136 (47.1%) were aged 60 years or older, with a mean age of 56 years and a median of 58 years. The sample comprised 132 (45.7%) male and 157 (54.3%) female participants, of whom the majority lived with at least one other person (248 [85.8%]).

**Media usage—Before quarantine.**   When asked about the type of media used before quarantine for obtaining information about the pandemic, the prevalent media outlet was television, as seen in Fig 2: 250 (86.5%; 95% CI [82.0, 90.2]) of the 289 participants reported occasional to very frequent use. This was followed by radio (197 [68.2%]; 95% CI [62.5, 73.5]), internet (153 [53.0%]; 95% CI [47.0, 58.8]) and partners/spouses (143 [49.5%]; 95% CI [43.6, 55.4]).

**Media usage—During quarantine.**   During quarantine, no significant changes in the types of media used by the participants were reported except for a slight, non-significant increase of internet use by 11 participants (3.8%) and information input from local authorities by 7 participants (2.4%), as well as a decrease in the use of newspapers reported by 31

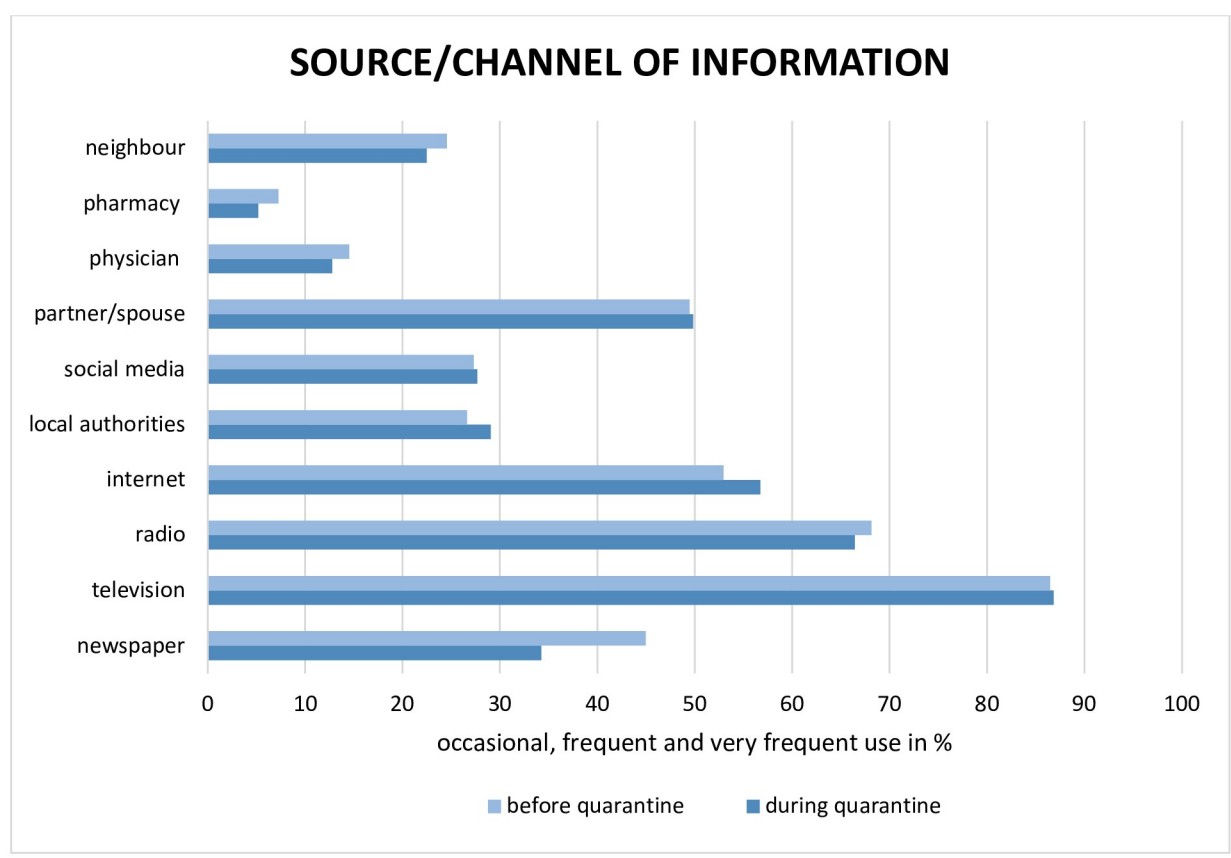

**Fig 2. Use of media and sources/channels of information before and during quarantine.** The graph shows the percentage of the combined results of occasional, frequent and very frequent use.

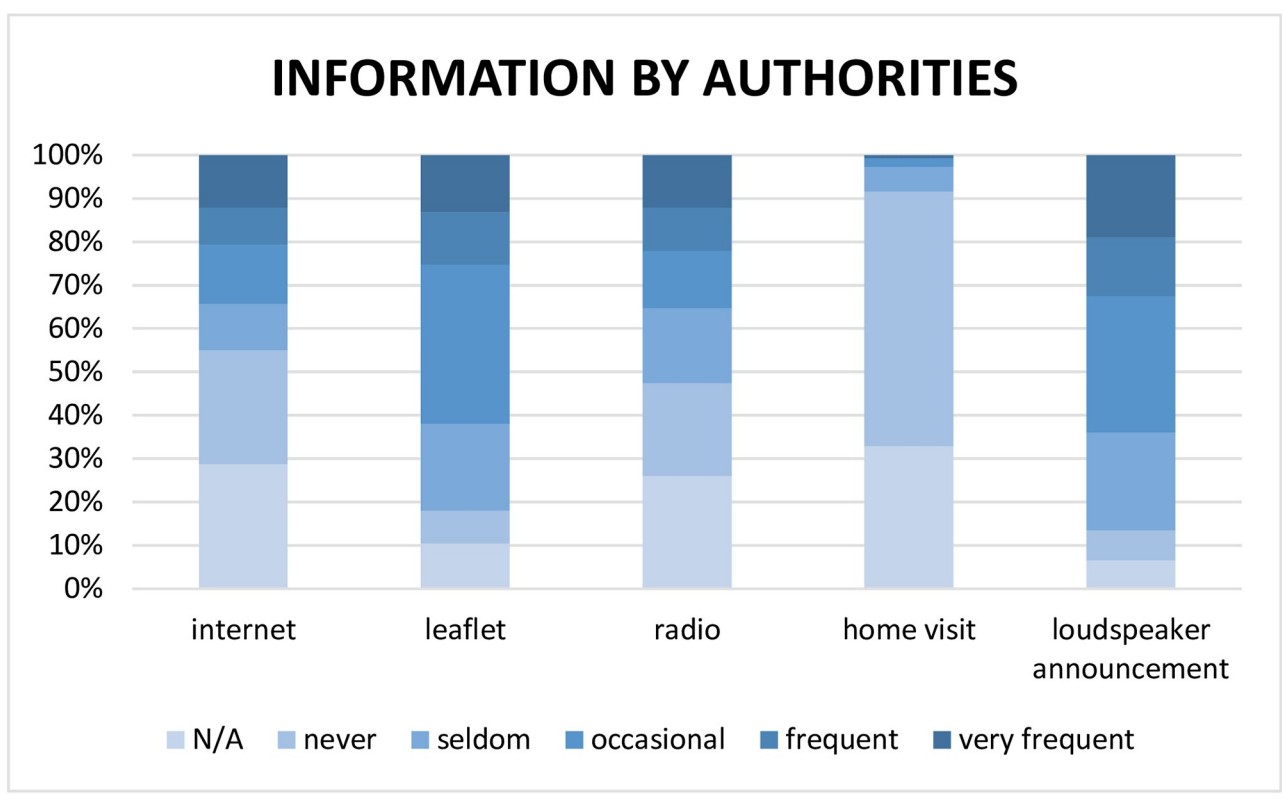

**Fig 3. Distribution of information by the local authorities.**

participants (10.7%). Television remained the most important channel of information both before and during quarantine, with 251 (86.8%; 95% CI [82.4, 90.5]) of the 289 participants reporting occasional to very frequent use.

With regard to the information obtained from the local authorities, participants primarily said they were occasionally, frequently and very frequently informed via leaflets (179 [61.9%]; 95% CI [56.1, 67.6]) or loudspeaker announcements (185 [64.0%]; 95% CI [58.2, 69.6]). Other means of communication (internet, radio, home visit) played only minor roles, as seen in Fig 3.

A moderate, good or very good level of information was reported by 211 participants (73.0%; 95% CI [67.5, 78.0]). Only 23 (8.0%; 95% CI [5.1, 11.7]) of the 289 participants said that there had been too much information. The level of concern during quarantine was reported to be moderate to very high by 200 (69.2%; 95% CI [63.5, 74.5]) of the 289 participants.

When asked about specific topics, the participants reported they were mostly concerned (at a moderate, high and very high level) about their families' health (249 [86.2%]; 95% CI [81.6, 89.9]), followed by their personal physical health (185 [64.0%]; 95% CI [58.2, 69.6]) and the nation's economic stability (183 [63.3%]; 95% CI [57.5, 68.9]). Personal mental health (154 [53.3%]; 95% CI [47.4, 59.2]), personal financial stability (153 [52.9%]; 95% CI [47.0, 58.8]) and the nation's political stability (152 [52.6%]; 95% CI [46.7, 58.5]) were all reported to be of a similar level of concern. Personal job security was not as relevant and was only reported as a concern by 105 participants (36.3%; 95% CI [30.8, 42.2]).

Regarding communication with the local authorities, the results show that most participants took the opportunity to speak directly to the authorities very infrequently or not at all. When contact was desired, the most established method of communication was a telephone hotline, which 68 participants (23.5%; 95% CI [18.8, 28.9]) reported having used occasionally, frequently or very frequently; 112 (38.8%; 95% CI [33.1, 44.6]) of the 289 participants responded that they had not sought any form of contact at any time.

Almost two-thirds of participants (212 [73.4%]; 95% CI [67.9, 78.4]) reported that the quarantine had been appropriate, one-third (68 [23.5%]; 95% CI [18.8, 28.9]) disagreed and nine (3.1%; 95% CI [1.4, 5.8]) did not give an answer. Two-thirds of respondents (217 [75.1%]; 95% CI [69.7, 80.0]) believed that the majority of the village's population had been compliant with the rules of the quarantine. When asked whether they could relate to people who did not comply with the rules, most participants (255 [88.2%]; 95% CI [83.9, 91.7]) articulated disapproval of non-compliance, and 116 (40.1%; 95% CI [34.4, 46.0]) said that they had avoided certain places or persons even after quarantine had ended.

## Explorative bivariate correlations

According to Spearman's rank correlation and the effect size classification by Cohen, we observed a weak correlation (0.17) between the items "level of information" and "level of concern" as seen in Table 1; that is, a higher level of information was associated with a high level of concern. An intermediate correlation (0.26) was found between the items "level of information" and "acceptance of quarantine", meaning a higher level of information correlates positively with an acceptance of the quarantine. Lastly, the items "level of concern" and "acceptance of quarantine" correlate weakly (0.16). Thus, a high level of concern correlates positively with an acceptance of the quarantine.

Fig 4 shows the degree of concern (from 1 [very low level of concern] to 5 [very high level of concern]) for each age group of participants. Participants between 60 and 79 years were the most concerned (average of 3.6). A similar observation was made for the self-reported level of information (from 1 [very poor level of information] to 5 [very good level of information]) by age group. For this item, participants who mostly thought themselves to be well informed (average of 3.6) were between 60 and 69 years old.

## Discussion

### Coordination and behaviour change

Our results show that 212 (73.4%) of the 289 participants agreed with the decision to impose community quarantine. This corresponds with the findings of the German COSMO study, which reported a hypothetical acceptance rate of 70% regarding a local lockdown for a representative sample of the German population in July 2020 [24].

The disapproval of quarantine non-compliance by 255 (88.2%) of the 289 respondents supports the finding of the high acceptance rate (212 [73.4%]) of the measure and underlines the participants' understanding of the importance of following the rules of the quarantine to contributing to the success of this intervention. This finding is representative of a population

**Table 1. Correlation between level of information, level of concern and acceptance of quarantine.**

|  | Spearman's ρ | p-value | n |
|---|---|---|---|
| level of information–level of concern | 0.17 | 0.005 | 275 |
| level of information–acceptance of quarantine | 0.26 | <0.001 | 273 |
| level of concern–acceptance of quarantine | 0.16 | 0.008 | 274 |

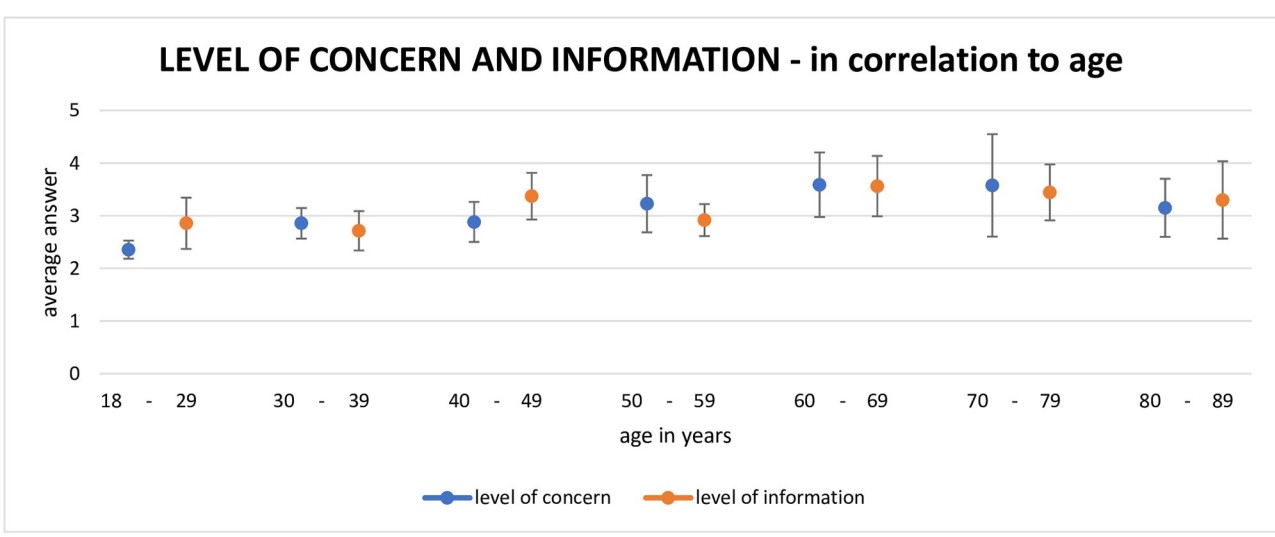

**Fig 4. Level of concern and level of information in correlation to age.** The graph shows the average answers of the participants according to their age group. 1 is no concern/very basic level of information, 5 is the highest level of concern/information.

whose age distribution represents a village society with a median age of 58, and thus many consider themselves at high risk for SARS-CoV-2. Furthermore, one should also keep in mind that the quarantine took place early on during the pandemic in Germany, when fear of contagion was high, thus promoting compliant behaviour [8].

The effects of quarantine were evident in the behaviour of the affected community well beyond the 14-day period: 116 (40.1%) of the 289 participants in our survey stated they continued to avoid places and people that they associated with the outbreak of infection even after the quarantine had been lifted. This is consistent with studies investigating earlier pandemics, such as that caused by the H1N1/09 virus during which behaviour changes were reported by 37.8% [28] and 68% [29] of study participants. The persistent behaviour change was not related to general guidance on personal hygiene and mask wearing; rather, it was connected to a place where an assumed super spreading event took place and to people that had been considered infectious.

## Information and communication

During quarantine, the same media channels were used as before, with national television being the most frequently mentioned channel [30], followed by radio. This may seem trivial, but stable and continuous media reception during an outbreak facilitates early and rapid risk communication with the population via appropriate channels. Surveys from 2019 reported that 77.1% of the German population listens to the radio on a daily basis, with an increase during times of crisis, as was also reported during the current pandemic [31,32].

The importance of radio as a media outlet and channel for trusted information has also been highlighted during other outbreaks. During the Ebola crisis in West Africa, radio community outreach programmes were important drivers of infection control, communication and community engagement [33–37]. The role of radio as a prime information channel seems to be consistent across low- and high-income countries during crises.

Internet and social media were not used as frequently as TV and radio, although there was a slight, non-significant increase during quarantine. One possible explanation for this lies in

the age distribution of Neustadt am Rennsteig. Compared to the participants under 60 years, the older participants reported a 22.2% and 30.7% lower use of the internet before and during quarantine, respectively. The difference regarding social media was 25.9% and 30.7%, respectively.

The decline in the use of newspapers as a channel of information could be attributed to the reported difficulties of postal delivery and delivery of press products at the beginning of quarantine, as mentioned in the open-ended responses (which were not analysed in this paper but will be published separately).

A total of 185 (64.0%) of the 289 participants reported loudspeaker announcements as a channel of information, especially in regard to locally relevant information. Loudspeaker announcements are associated with the pre-smartphone age but are still occasionally used in rural areas.

Moreover, partners or spouses played a more important role in sharing information than professionals (*e.g.* pharmacists, general practitioners or local authorities), pointing towards a rather horizontal flow of information across the same level. This is supported by the data showing that there was scarce contact with the authorities from part of the population. Interestingly, participants reported only a slight, non-significant increase in the (low-level) importance of local authorities as sources of information during the quarantine.

This suggests that health authorities might not be the primary or most trusted sources of information for the community. Considering the key role public health authorities have in the management of a public health crisis, fostering rapport and engaging with communities even before a public health event would be worthwhile [6].

The reported good level of information (73.0%) regarding SARS-CoV-2 and the reported acceptance of the community quarantine (73.4%) mirror the findings of other research in the German population [24,38,39]. This is especially important because a good level of information–as well as a high level of concern–correlate positively with a high acceptance of community quarantine and high reported compliance.

Therefore, a good flow of information about the need for and reasons behind quarantine may significantly influence its acceptance. This corresponds with previous findings regarding the influence of effective risk communication on the population's behaviour as researched during the SARS pandemic in 2003 [10].

The level of information correlates with the level of concern, meaning that people who reported a high level of information had a high level of concern and vice versa. This was particularly true for the older population (over the age of 60), who were addressed in the general mass communication as "high-risk groups".

Investigating the level of concern further, we found that the participants were mostly worried about their families' health, even more than their own health. This could be based on the so-called optimism bias, which leads to a greater perceived risk for others than for oneself [7]. Interestingly, employment security was only a concern for a small proportion of the participants, indicating the age and work biographies of the participants. However, this finding is surprisingly consistent with other surveys that reported an even lower level of concern, around 20%, during recent months [24]. The reasons for this should be further researched but could indicate a high level of trust in the German government to secure workplaces and support the economy. Research has shown that trust in the national government was at a high of 45% in July 2020 [40].

## Limitations

Regarding the reported acceptance of and compliance with the quarantine, we have no knowledge if our results are specific to this particular selection of participants. It is possible that only

those who approved of the intervention participated in the study, thus distorting results via selection bias. This means that the results may not represent the opinion and views of the population of Neustadt am Rennsteig as a whole but rather are representative for a specific part of the population who was willing to participate in the study. Moreover, we cannot exclude social desirability bias as the questionnaire was distributed in connection with the CoNAN seroprevalence study. Villagers who were ashamed of their (actual or presumed) previous SARS-Cov-2 infection might have not taken part in the seroprevalence study, therefore not being eligible for the risk communication study.

Additionally, resistance against quarantine orders often originates in individuals who are around the age of 30, a group that is underrepresented in our sample [41]. This age group's information requirements should therefore be assessed in future studies.

As the survey took place six weeks after the end of the community quarantine, we also cannot exclude memory bias.

Considering these limitations, the results of our study are specific for a population which is defined by certain characteristics, for example mean age of 56 years and residency in a small German village. This results in a limited transferability of our conclusions onto the broader public of Germany.

## Conclusion

This study is one of the first to investigate the closely defined cluster of a whole village under quarantine. It gives a short, quantified summary of the affected population's reception of the risk communication, information and coordination during a community quarantine order in Germany in the context of the SARS-CoV-2 pandemic.

Our research shows that there was very little change in the channels of information used during quarantine, with television remaining the most important one. Moreover, there was only limited contact with the local authorities, which we interpreted as a sign that horizontal information exchange among peers was more important. In order to design and implement successful risk communication strategies, it is important to acknowledge the preferred channels of information and media outlets of the community.

An important finding of our research is the reported high approval of the community quarantine, which was significantly associated with information level. This shows that compliance with quarantine can be encouraged and improved by having adequate information routines in place. Public health authorities should increase their impact by improving community engagement and communication.

The quarantine in Neustadt was a success in terms of the population's acceptance and support of the measure. The quarantine also contributed to limiting and ending the disease transmission in this village. Keeping in mind the assumed limitations and therefore possibly restricted generalisation of our results, these findings could contribute to a framework for public health risk communication in order to facilitate effective public health interventions.

## Supporting information

**S1 Table. Results in detail.** The data of the questionnaire are listed in detail (whole numbers and percent) for every question, except for the open-ended questions.
(PDF)

**S1 File. Questionnaire.** The questionnaire which was used to obtain the data.
(PDF)

**S2 File. Information leaflet.** The information leaflet was given out to the population of Neustadt am Rennsteig during quarantine. It explains how postal deliveries, waste disposal and the working situation were handled.
(PDF)

**S3 File. Issued order.** The order contains the announcement of quarantine in Neustadt am Rennsteig. It explains who is affected by quarantine and threatens with consequences in case of breaking the rules.
(PDF)

## Acknowledgments

We wish to extend our appreciation to the citizens of Neustadt am Rennsteig who participated in our study.

### Study group

Technische Universität Ilmenau, Ilmenau, Germany: Thomas Hotz

Local Cooperation partners: Petra Enders, Renate Koch, Steffen Mai, Matthias Ullrich

Institute of Clinical Chemistry and Laboratory Diagnostics and Integrated Biobank Jena (IBBJ), Jena University Hospital–Friedrich Schiller University, Jena, Germany: Cora Richert, Cornelius Eibner, Bettina Meinung, Kay Stötzer, Julia Köhler, Michael Kiehntopf, Dagmar Rimek

Children's Hospital, Jena University Hospital–Friedrich Schiller University, Jena, Germany: Hans Cipowicz, Christine Pinkwart, Hans Proquitté

Institute for Infectious Disease and Infection Control, Jena University Hospital–Friedrich Schiller University, Jena, Germany: Steffi Kolanos, Juliane Ankert, Oliwia Makarewicz, Stefan Hagel, Christina Bahrs, Aurelia Kimmig, Anita Hartung, Daniel Weiss, Lara Thieme, Gabi Hanf, Clara Schnizer, Jasmin Müller, Jennifer Kosenkow, Franziska Röstel

Institute of Immunology, Jena University Hospital–Friedrich Schiller University, Jena, Germany: Nico Andreas, Raphaela Marquardt, Thomas Kamradt

Institute of Medical Microbiology, Jena University Hospital–Friedrich Schiller University, Jena, Germany: Stefanie Deinhardt-Emmer, Sebastian Kuhn, Stefan Glöckner, Michael Baier, Bettina Löffler

Department of Otorhinolaryngology, Jena University Hospital–Friedrich Schiller University, Jena, Germany: Hilmar Gudziol, Timo Kirschstein, Orlando Guntinas-Lichius, Thomas Bitter

Center for Sepsis Control and Care (CSCC)–Friedrich Schiller University, Jena, Germany: Joel Guerra

### Lead authors

Mathias W. Pletz, MD; Sebastian Weis, MD

Mail: Mathias.Pletz@med.uni-jena.de; Sebastian.Weis@med.uni-jena.de

## Author Contributions

**Conceptualization:** Juliane Scholz, Wibke Wetzker, Annika Licht, Petra Dickmann.

**Data curation:** Juliane Scholz, Annika Licht, Sebastian Weis, Mathias Pletz.

**Formal analysis:** Juliane Scholz, Rainer Heintzmann, André Scherag.

**Funding acquisition:** Petra Dickmann.

**Investigation:** Juliane Scholz, Wibke Wetzker, Annika Licht, Petra Dickmann.

**Methodology:** Juliane Scholz, Rainer Heintzmann, André Scherag.

**Project administration:** Petra Dickmann.

**Resources:** Sebastian Weis, Mathias Pletz, Michael Bauer.

**Supervision:** Wibke Wetzker, Petra Dickmann.

**Validation:** Wibke Wetzker, Rainer Heintzmann, Cornelia Betsch, Michael Bauer.

**Visualization:** Juliane Scholz, Rainer Heintzmann.

**Writing – original draft:** Juliane Scholz.

**Writing – review & editing:** Wibke Wetzker, Annika Licht, Rainer Heintzmann, André Scherag, Sebastian Weis, Mathias Pletz, Cornelia Betsch, Michael Bauer, Petra Dickmann.

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
