## [Decision Letter · Decision Letter 0]

10 May 2021

PONE-D-21-02054

The role of risk communication in public health interventions. An analysis of risk communication for a community quarantine in Germany to curb the SARS-CoV-2 pandemic.

PLOS ONE

Dear Dr. Scholz,

Thank you for submitting your manuscript to PLOS ONE. After careful consideration, we feel that it has merit but does not fully meet PLOS ONE’s publication criteria as it currently stands. Therefore, we invite you to submit a revised version of the manuscript that addresses the points raised during the review process.

Please provide further details as requested from reviewer 1. Please address the issue of bias as raised by reviewer 2.;

We look forward to receiving your revised manuscript.

Kind regards,

Rosemary Frey

Academic Editor

PLOS ONE

Journal Requirements:

3) In the Methods section, please provide additional details regarding the questionnaire/ survey validation.

4) We note that Supplementary Material (Information Leaflet) in your submission may contain copyrighted images. All PLOS content is published under the Creative Commons Attribution License (CC BY 4.0), which means that the manuscript, images, and Supporting Information files will be freely available online, and any third party is permitted to access, download, copy, distribute, and use these materials in any way, even commercially, with proper attribution. For more information, see our copyright guidelines: http://journals.plos.org/plosone/s/licenses-and-copyright.

i.         You may seek permission from the original copyright holder of Figure(s) [#] to publish the content specifically under the CC BY 4.0 license.

ii.    If you are unable to obtain permission from the original copyright holder to publish these figures under the CC BY 4.0 license or if the copyright holder’s requirements are incompatible with the CC BY 4.0 license, please either i) remove the figure or ii) supply a replacement figure that complies with the CC BY 4.0 license. Please check copyright information on all replacement figures and update the figure caption with source information. If applicable, please specify in the figure caption text when a figure is similar but not identical to the original image and is therefore for illustrative purposes only.

5) Your ethics statement should only appear in the Methods section of your manuscript. If your ethics statement is written in any section besides the Methods, please move it to the Methods section and delete it from any other section. Please ensure that your ethics statement is included in your manuscript, as the ethics statement entered into the online submission form will not be published alongside your manuscript.

6) We note you have included a table to which you do not refer in the text of your manuscript. Please ensure that you refer to Table 1 in your text; if accepted, production will need this reference to link the reader to the Table.

7) Please include captions for your Supporting Information files at the end of your manuscript, and update any in-text citations to match accordingly. Please see our Supporting Information guidelines for more information: http://journals.plos.org/plosone/s/supporting-information.

8) One of the noted authors is a group or consortium - CoNAN study group. In addition to naming the author group in the acknowledgements, please also indicate clearly a lead author for this group along with a contact email address.

Reviewers' comments:

Reviewer's Responses to Questions

**Comments to the Author**

1. Is the manuscript technically sound, and do the data support the conclusions?

Reviewer #1: Yes

Reviewer #2: Partly

2. Has the statistical analysis been performed appropriately and rigorously? 

Reviewer #1: Yes

Reviewer #2: Yes

3. Have the authors made all data underlying the findings in their manuscript fully available?

Reviewer #1: Yes

Reviewer #2: Yes

4. Is the manuscript presented in an intelligible fashion and written in standard English?

Reviewer #1: Yes

Reviewer #2: Yes

5. Review Comments to the Author

Reviewer #1: The authors present an interesting case study of risk communication examining a community quarantine in a German town. The paper presents only the quantitative aspects of the survey they fielded, qualitative results will follow in another planned manuscript. The article is well-written but perhaps too concise. The findings reflect important aspects of communication that will be useful to public health authorities in planning and implementing future interventions.

I think the paper would benefit from some additional background and clarity of terms, as listed below:

Pg 4, Background

I would like more explanation of the quarantine implementation and the related communication effort. Who was responsible, was it the local health authority? What protocol did they follow, something designed locally or other system (EU or WHO)? Similarly, was the communication effort designed locally or a pre-existing plan?

Pg 5, Methods

I would like more detail on the concept/definition of 'level of information'. This seems to be key to the whole project of good communication since 'level of information' was statistically significantly correlated with 'acceptance of quarantine'. Getting an affected community the right "level of information" would be a goal for future communication efforts.

Because terms like 'source' of communication are used in different ways I suggest the authors include in the Methods section a simple model of communication (Berlo's Source-Message-Channel-Receiver might work) to add clarity. For example, on page 11 (line 334), loud speaker announcements are called a 'source' and then later on there is discussion of whether public health authorities are a trusted 'source' (line 345). Using the Berlo model, public health authorities are a source and loud speaker announcements are considered a channel.

Pg 6 line 153: Please include description of how was the survey returned.

Reviewer #2: The social health importance and timeliness of this research are obvious and commendable.

The likely biases in the sample (selection bias, age bias, social desirability bias) matter a great deal to the viability of this study however, and this should be more thoroughly acknowledged at relevant points throughout the paper. Those biases make it difficult to draw very accurate conclusions about the wider population with respect to the descriptive statistics cited, and even moreso with the inferential conclusions drawn from the correlations between variables, and associated p values. How for example do you know the reported 'high acceptance of quarantine', 'information levels', and 'media exposure patterns' actually reflect the Neustadt am Rennsteig population as a whole--if certain types of respondents have been underrepresented? Given the strong views of those who oppose Covid community interventions, it seems unlikely that survey nonresponse was just randomly distributed, with no appreciable affect on the results.

To the authors' credit, the 'implications' section does briefly acknowledge the likelihood that those not supporting the intervention might have also declined participation in this study, and also the unevenness in study participation across the age spectrum. However this weakness needs to be more thoroughly discussed, and the limitations in what can, and cannot, be concluded--given the sample limitations--more clearly spelled out for the reader. As it presently stands, the paper appears to be drawing more extensive conclusions about the subject than are warranted given the likely skewed sample. The fact that the "questionnaire was handed out together with the epidemiological questionnaire" also suggests there may have been some 'social desirability bias' affecting the question responses. This, too, should be considered in the limitations section.

Minor issue: Grammatically in English, 'data' are plural, not singular, so make sure you say 'data are' rather than 'data is'. (in line 156 for example)

6. PLOS authors have the option to publish the peer review history of their article (what does this mean?). If published, this will include your full peer review and any attached files.

Reviewer #1: No

Reviewer #2: No

---

## [Author Response · Author response to Decision Letter 0]

8 Jun 2021

Dear editor and reviewers, 

thank you very much for your time and helpful feedback. I have taken all mentioned points into consideration and revised the manuscript accordingly. 

One reference has been changed because it was cited as a preprint at the time of writing but since then has been published. 

[1] 1. Streeck H, Schulte B, Kümmerer BM, Richter E, Höller T, Fuhrmann C, et al. Infection fatality rate of SARS-CoV2 in a super-spreading event in Germany. Nature Communications. 2020;11(1):5829.

File naming has been adapted to meet the PLOS ONE’s style requirements. Hopefully, everything can now be found in accordance to the style. 

3) In the Methods section, please provide additional details regarding the questionnaire/ survey validation.

Survey validation has been discussed more in detail und it is explained that due to the urgent need for scientific information accompanied by time constraints at the beginning of the pandemic, there was only limited time to develop a thorough study design and that we had to focus on efficiency and speediness. Moreover, due to the small study population, we were reluctant to pilot the questionnaire within the population of Neustadt am Rennsteig as this would mean the loss of a significant portion of potential participants. 

4) We note that Supplementary Material (Information Leaflet) in your submission may contain copyrighted images. All PLOS content is published under the Creative Commons Attribution License (CC BY 4.0), which means that the manuscript, images, and Supporting Information files will be freely available online, and any third party is permitted to access, download, copy, distribute, and use these materials in any way, even commercially, with proper attribution. For more information, see our copyright guidelines: http://journals.plos.org/plosone/s/licenses-and-copyright.

We require you to either (1) present written permission from the copyright holder to publish these figures specifically under the CC BY 4.0 license, or (2) remove the figures from your submission. 

The image in question has been removed and the Information Leaflet has been resubmitted. 

5) Your ethics statement should only appear in the Methods section of your manuscript. If your ethics statement is written in any section besides the Methods, please move it to the Methods section and delete it from any other section. Please ensure that your ethics statement is included in your manuscript, as the ethics statement entered into the online submission form will not be published alongside your manuscript.

The ethics statement has been moved to the Methods section. 

6) We note you have included a table to which you do not refer in the text of your manuscript. Please ensure that you refer to Table 1 in your text; if accepted, production will need this reference to link the reader to the Table.

Table 1 is now mentioned in the text. 

7) Please include captions for your Supporting Information files at the end of your manuscript, and update any in-text citations to match accordingly. Please see our Supporting Information guidelines for more information: http://journals.plos.org/plosone/s/supporting-information.

Captions to the Supporting files have been added. 

8) One of the noted authors is a group or consortium - CoNAN study group. In addition to naming the author group in the acknowledgements, please also indicate clearly a lead author for this group along with a contact email address.

The two lead authors of the CoNAN study group have been added with contact email addresses. 

Reviewer #1: 

Pg 4, Background

I would like more explanation of the quarantine implementation and the related communication effort. Who was responsible, was it the local health authority? What protocol did they follow, something designed locally or other system (EU or WHO)? Similarly, was the communication effort designed locally or a pre-existing plan?

Responsible for the implementation of and communication during community quarantine were the local health authorities. They did not follow a pre-designed protocol but drafted an ad-hoc plan, changing their strategy day by day, adapting to upcoming issues.

Pg 5, Methods

I would like more detail on the concept/definition of 'level of information'. This seems to be key to the whole project of good communication since 'level of information' was statistically significantly correlated with 'acceptance of quarantine'. Getting an affected community the right "level of information" would be a goal for future communication efforts. 

The structure of the survey was based on the understanding of risk communication as being composed by three pillars: information, communication and coordination. All three components contribute to capacity building and preparing for the event of a public health crisis. The importance of information in the context of a public health emergency is showcased by the definition of risk communication as “information exchange about health risks caused by environment, industrial, or agricultural, processes, policies, or products among individuals, groups and institutions”. Therefore, the participants’ self-reported level of information was one of our focal points. In our study, level of information was defined as the participants’ personal grasp of quarantine and the reasons behind the decision which is closely related to the sources and channels of information which were used to access information and which we strived to identify.

Because terms like 'source' of communication are used in different ways I suggest the authors include in the Methods section a simple model of communication (Berlo's Source-Message-Channel-Receiver might work) to add clarity. For example, on page 11 (line 334), loud speaker announcements are called a 'source' and then later on there is discussion of whether public health authorities are a trusted 'source' (line 345). Using the Berlo model, public health authorities are a source and loud speaker announcements are considered a channel.

I have incorporated the model of Berlo into the Methods section and changed the corresponding text passages accordingly. 

Pg 6 line 153: Please include description of how was the survey returned.

A description of how the surveys returned has been added. 

Reviewer #2: 

The social health importance and timeliness of this research are obvious and commendable.

The likely biases in the sample (selection bias, age bias, social desirability bias) matter a great deal to the viability of this study however, and this should be more thoroughly acknowledged at relevant points throughout the paper. Those biases make it difficult to draw very accurate conclusions about the wider population with respect to the descriptive statistics cited, and even moreso with the inferential conclusions drawn from the correlations between variables, and associated p values. How for example do you know the reported 'high acceptance of quarantine', 'information levels', and 'media exposure patterns' actually reflect the Neustadt am Rennsteig population as a whole--if certain types of respondents have been underrepresented? Given the strong views of those who oppose Covid community interventions, it seems unlikely that survey nonresponse was just randomly distributed, with no appreciable affect on the results.

To the authors' credit, the 'implications' section does briefly acknowledge the likelihood that those not supporting the intervention might have also declined participation in this study, and also the unevenness in study participation across the age spectrum. However this weakness needs to be more thoroughly discussed, and the limitations in what can, and cannot, be concluded--given the sample limitations--more clearly spelled out for the reader. As it presently stands, the paper appears to be drawing more extensive conclusions about the subject than are warranted given the likely skewed sample. The fact that the "questionnaire was handed out together with the epidemiological questionnaire" also suggests there may have been some 'social desirability bias' affecting the question responses. This, too, should be considered in the limitations section.

The topic of biases is now being more thoroughly discussed in the limitations and has also been incorporated in the conclusions, so hopefully the reader can gain a clearer picture of our study’s results. 

Minor issue: Grammatically in English, 'data' are plural, not singular, so make sure you say 'data are' rather than 'data is'. (in line 156 for example)

I have changed “is” to “are” when talking about data.

Thank you again for your comments. I hope I could clarify any questions and have incorporated all of your suggestions accordingly. 

Kind regards, 

Juliane Scholz

---

## [Decision Letter · Decision Letter 1]

2 Aug 2021

The role of risk communication in public health interventions. An analysis of risk communication for a community quarantine in Germany to curb the SARS-CoV-2 pandemic.

PONE-D-21-02054R1

Dear Dr. Scholz,

We’re pleased to inform you that your manuscript has been judged scientifically suitable for publication and will be formally accepted for publication once it meets all outstanding technical requirements.

Kind regards,

Rosemary Frey

Academic Editor

PLOS ONE

Additional Editor Comments (optional):

Reviewers' comments:

Reviewer's Responses to Questions

**Comments to the Author**

1. If the authors have adequately addressed your comments raised in a previous round of review and you feel that this manuscript is now acceptable for publication, you may indicate that here to bypass the “Comments to the Author” section, enter your conflict of interest statement in the “Confidential to Editor” section, and submit your "Accept" recommendation.

Reviewer #1: All comments have been addressed

Reviewer #2: All comments have been addressed

2. Is the manuscript technically sound, and do the data support the conclusions?

Reviewer #1: Yes

Reviewer #2: Partly

3. Has the statistical analysis been performed appropriately and rigorously? 

Reviewer #1: Yes

Reviewer #2: Yes

4. Have the authors made all data underlying the findings in their manuscript fully available?

Reviewer #1: Yes

Reviewer #2: Yes

5. Is the manuscript presented in an intelligible fashion and written in standard English?

Reviewer #1: Yes

Reviewer #2: Yes

6. Review Comments to the Author

Reviewer #1: (No Response)

Reviewer #2: The authors' thorough discussion in the 'limitations' section of the likely selection, age and social desirability biases, which limit the generalizability of the study results, now makes the presentation more balanced. Although some of these factors should have been foreseen at the stage of initial study design, not after completion, this study nevertheless makes a useful contribution and should be published.

7. PLOS authors have the option to publish the peer review history of their article (what does this mean?). If published, this will include your full peer review and any attached files.

Reviewer #1: No

Reviewer #2: No

---

## [Editor Report · Acceptance letter]

4 Aug 2021

PONE-D-21-02054R1 

The role of risk communication in public health interventions. An analysis of risk communication for a community quarantine in Germany to curb the SARS-CoV-2 pandemic. 

Dear Dr. Scholz:

I'm pleased to inform you that your manuscript has been deemed suitable for publication in PLOS ONE. Congratulations! Your manuscript is now with our production department. 

Kind regards, 

on behalf of

Dr. Rosemary Frey 

Academic Editor

PLOS ONE